# Learning to Drop Out: An Adversarial Approach to Training Sequence VAEs

**Đorđe Miladinović** [*†]   **Kumar Shridhar** [*‡]   **Kushal Jain** [¶]
**Max B. Paulus** [‡]   **Joachim M. Buhmann** [‡]   **Mrinmaya Sachan** [‡]   **Carl Allen** [‡]

[†]GSK.ai   [‡]ETH Zürich   [¶]University of California, San Diego

## Abstract

In principle, applying variational autoencoders (VAEs) to sequential data offers a method for controlled sequence generation, manipulation, and structured representation learning. However, training sequence VAEs is challenging: autoregressive decoders can often explain the data without utilizing the latent space, known as *posterior collapse*. To mitigate this, state-of-the-art models 'weaken' the 'powerful' decoder by applying uniformly random *dropout* to the decoder input. We show theoretically that this removes *pointwise mutual information* provided by the decoder input, which is compensated for by utilizing the latent space. We then propose an *adversarial* training strategy to achieve *information-based stochastic dropout*. Compared to uniform dropout on standard text benchmark datasets, our targeted approach increases both sequence modeling performance and the information captured in the latent space.

## 1   Introduction

Training autoregressive models via maximum likelihood estimation (MLE) is a common strategy for representing sequential data. Such autoregressive models obtain state-of-the-art in modeling text [Brown et al., 2020, Chowdhery et al., 2022, Zhang et al., 2022], speech [Oord et al., 2018, Conneau et al., 2021] , and video sequences [Babaeizadeh et al., 2018]. In this simple and intuitive approach, the joint distribution of a sequence is factorized into a product of conditional distributions, with each sequence element conditioned on its history. However, in their basic form, autoregressive models do not necessarily learn *latent variables* that encode informative content, as often desired. As such, supplementing autoregressive models with latent variables is a promising way to enhance control in sequence generation, and enable structured representation learning.

A recent approach to latent variable modeling of sequences is to integrate autoregressive components into the *variational autoencoder* (VAE) framework [Kingma and Welling, 2014, Rezende et al., 2014]. *Sequence VAEs* offer a theoretically principled solution, but have not been widely adopted. Arguably, this is largely due to the phenomenon of *posterior collapse* [Bowman et al., 2016, Chen et al., 2017, Van Den Oord et al., 2017, Chen et al., 2020]. Posterior collapse describes when a VAE's posterior probability over latent variables 'collapses' to the latent prior, rendering the latent space completely uninformative, or *independent*, of the data. Recent years have seen an abundance of research into posterior collapse [Bowman et al., 2015, Chen et al., 2017, Lucas et al., 2019, Dai et al., 2020, Fang et al., 2021, Pang et al., 2021], identifying the 'power' of autoregressive decoding as a main cause. In particular, autoregressive decoders are shown to obtain satisfactory sequence modeling performance without utilizing the latent space [Bowman et al., 2016, Chen et al., 2017]. This suggests that for sequence VAEs to progress, improved techniques are needed to understand and alleviate posterior

---

[*]Equal contribution; correspondence at: shkumar@ethz.ch

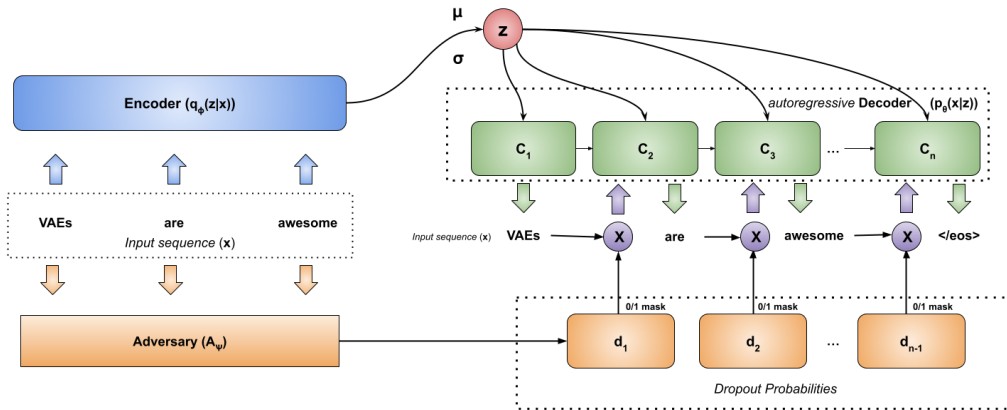

Figure 1: Adversarial training of sequence VAEs. Our proposed model comprises **an encoder**, **an autoregressive sequence decoder** (together an autoregressive VAE), and **an adversary $A_\psi$**. During training, the encoder learns a representation of a full input sequence $x$; the same sequence serves as both input (together with the hidden representation) and target to train the decoder. The adversary *learns to stochastically drop out* sequence elements that the decoder requires most, **masking (0/1)** each sequence element $x_i$ with **probability $d_i$**. See Section 4.2 for details.

collapse, and thereby learn informative latent structure. In this work, we propose a theoretically principled way to mitigate posterior collapse when training autoregressive sequence VAEs.

The intuition that autoregressive VAE decoders are too 'powerful' has led to the notion of 'weakening' them, e.g. by regularization. However, weakening an autoregressive model arbitrarily might of course harm its performance, leading to a trade off between informativeness of the latent space and the quality of sequence modeling (or density estimation performance). State-of-the-art VAEs apply variants of *dropout* [Srivastava et al., 2014] to the sequence input into the decoder: either to individual dimensions of sequence elements [Kim et al., 2018], or to mask entire sequence elements [Iyyer et al., 2015, Bowman et al., 2016]. In either case, dropout is generally applied uniformly at random, meaning that each sequence element has an equal probability of being dropped. From the intuition that different words carry different information about the next word, this work investigates whether a *non-uniform* dropout policy can achieve a better trade-off between capturing latent information and sequence modeling performance. Specifically, by stochastically dropping each sequence element according to its importance to the autoregressive decoder, our method takes a *targeted* approach to dropout relative to uniform sampling, maintaining sequence modeling performance while achieving the benefits of dropout on the latent space. To apply a non-uniform, data-dependent dropout scheme, our approach introduces an 'adversary' that *learns which elements to drop*. The proposed framework is represented in Figure 1, theoretically developed in Section 4.1 and implemented in Section 4.2.

The contributions of this paper can be summarized as follows: **(i)** We propose an adversarial approach to learning a stochastic dropout policy that mitigates posterior collapse in sequence VAEs. **(ii)** We show theoretically that dropping out sequence elements deducts *mutual information not already learned by the model*, between one sequence element and the next, and that our approach maximises this quantity, leading to a constrained minimax ELBO objective that explains an adversarial approach. **(iii)** We evaluate the proposed scheme on standard text benchmarks, showing that our approach renders an informative latent space, without trading-off but rather improving sentence modeling. **(iv)** We examine properties of the adversarial network and find that the adversary typically selects words that carry sentence semantics that fit our theoretical analysis.

## 2 Background

Here, we formally describe the use of autoregressive decoding within VAEs for sequential data and the problem of *posterior collapse* that often arises (refer to Appendix D for some clarification around the term *posterior collapse*).

For notation: $\boldsymbol{X}$ denotes a random variable, (boldface) $\boldsymbol{x}$ denotes a *sequence* that is a realization of $\boldsymbol{X}$, (subscript) $\boldsymbol{x}_i$ denotes the $i$-th sequence element of $\boldsymbol{x}$ and $\boldsymbol{x}_{<i}$ collectively denotes all sequence elements prior $\boldsymbol{x}_i$. $z$ denotes a global latent vector and (superscript) $\boldsymbol{x}^{(i)}$ denotes the $i$-th sample from a dataset.

The task is to approximate the true probability density function $p_*(\boldsymbol{x})$ of a random sequence $\boldsymbol{X} \sim p_*(\boldsymbol{x})$ using a model $p_\theta(\boldsymbol{x})$ with parameters $\theta$. Given a set of observations $\{\boldsymbol{x}^{(i)}\}_{i=1}^N$, $\theta$ can be estimated by *maximum likelihood estimation* (MLE):

$$\theta_{MLE} = \underset{\theta}{\operatorname{argmax}} \left[ \frac{1}{N} \sum_{i=1}^N \log p_\theta(\boldsymbol{x}^{(i)}) \right] \tag{1}$$

If the data has latent generative factors $z$, modelled as $p_\theta(\boldsymbol{x}) = \int_z p_\theta(\boldsymbol{x}|z)p_\theta(z)$, Eq 1 is intractable and one can instead maximize the *evidence lower bound* (ELBO), which for any observation $\boldsymbol{x}$ is:

$$\log p(\boldsymbol{x}) \geq \log p_\theta(\boldsymbol{x}) \geq \text{ELBO}_{\theta,\phi}(\boldsymbol{x}) \triangleq \int_z q_\phi(z|\boldsymbol{x}) \log p_\theta(\boldsymbol{x}|z) - \int_z q_\phi(z|\boldsymbol{x}) \log \frac{q_\phi(z|\boldsymbol{x})}{p_\theta(z)} \tag{2}$$

where, $q_\phi(z|\boldsymbol{x})$ approximates the posterior, parameterized by $\phi$, and $p_\theta(z)$ is a prior distribution over latent variables. In this work, we consider the framework of variational autoencoders (VAEs) [Kingma and Welling, 2014, Rezende et al., 2014], where the *encoder* $q_\phi(z|\boldsymbol{x})$ and *decoder* $p_\theta(\boldsymbol{x}|z)$ are parameterised by neural networks and $p_\theta(z)$ is a standard Gaussian, denoted $p_0(z)$. Further, we focus specifically on VAEs where the decoder $p_\theta(\boldsymbol{x}|z)$ is *autoregressive* [Bowman et al., 2016, Chen et al., 2020, He et al., 2019], meaning $\log p_\theta(\boldsymbol{x}|z) = \sum_i \log p_\theta(x_i|\boldsymbol{x}_{<i}, z)$, and the ELBO becomes:

$$\text{ELBO}_{\theta,\phi}^{\textbf{AR}}(\boldsymbol{x}) = \int_z q_\phi(z|\boldsymbol{x}) \sum_i \log p_\theta(\boldsymbol{x}_i|\boldsymbol{x}_{<i}, z) - \underbrace{\int_z q_\phi(z|\boldsymbol{x}) \log \frac{q_\phi(z|\boldsymbol{x})}{p_0(z)}}_{\text{KL}} \tag{3}$$

**Posterior collapse:** Autoregressive models improve density estimation [Oord et al., 2016] by explicitly modeling statistical dependencies between sequence elements. However, when used as a VAE decoder $p_\theta(\boldsymbol{x}|z)$, autoregressive models are often found to ignore $z$, i.e. $p(\boldsymbol{x}|z) \approx p(\boldsymbol{x})$, and the posterior is said to 'collapse' to the prior, i.e. $q_\theta(z|\boldsymbol{x}) \approx p_0(z)$, rendering $z$ *de facto* independent of $\boldsymbol{x}$ [Bowman et al., 2016, He et al., 2019]. In terms of Eq 3, this implies $p_\theta(\boldsymbol{x}_i|\boldsymbol{x}_{<i}, z) \approx p_\theta(\boldsymbol{x}_i|\boldsymbol{x}_{<i})$. From this, posterior collapse can be considered due to the *information* about $\boldsymbol{x}_i$ provided by $\boldsymbol{x}_{<i}$ being such that each $\boldsymbol{x}_i$ is (approximately) conditionally independent of $z$ given $\boldsymbol{x}_{<i}$, in other words $z$ is essentially *redundant*. Based on this, we interpret the notion that autoregressive models are *too powerful* for use as a VAE decoder to mean that they allow *too much information flow*.

## 3  Related Work

**Posterior collapse**   The phenomenon of posterior collapse has been observed in the context of text [Bowman et al., 2016, Yang et al., 2017], images [Chen et al., 2017, Razavi et al., 2018, Miladinović et al., 2021], videos [Babaeizadeh et al., 2018, Miladinović et al., 2019a], speech [Chorowski et al., 2019] and graphs [Kipf et al., 2018]. In broad terms, previous solutions can be divided into two complementary categories: **(i)** *latent-variable-oriented* methods that focus mainly on relaxing the KL penalty; and **(ii)** *decoder-oriented* methods that regularise or 'weaken' the autoregressive decoder. Typically, both types are required in order to learn an informative latent space [Bowman et al., 2016, Chen et al., 2017, Kim et al., 2018].

**Latent-variable-oriented solutions**  Bowman et al. [2016] suggest *KL annealing*, a technique that introduces the KL term gradually into training according to a predefined schedule. *Free bits* [Kingma et al., 2016] prevent penalizing of the KL term if its magnitude is below a predefined threshold. Van Den Oord et al. [2017] use a *discrete* VAE, avoiding posterior collapse by design. Fu et al. [2019] apply a *cyclical* form of KL annealing. Razavi et al. [2018] constrain the variational family of the posterior distribution (encoder), preventing it from closely approximating the prior and so holding the KL term away from zero. *Lagging inference networks* [He et al., 2019] 'aggressively' optimize the

encoder before each decoder update. A similar procedure is followed by *Semi-Amortized VAEs* [Kim et al., 2018]. *Generative skip models* [Dieng et al., 2019] introduce skip connections to create a more explicit link between the latent variables and the likelihood function. Sinha and Dieng [2021] propose a *consistency* regularizer by minimizing the KL divergence between the posterior approximations of an observation and a random transformation of it.

**Decoder-oriented solutions** Bowman et al. [2016] apply *word dropout* [Iyyer et al., 2015] to uniformly drop words during autoregressive decoding. Other methods [Kim et al., 2018, He et al., 2019] apply parameter dropout [Srivastava et al., 2014] to word embeddings. Chen et al. [2017], Semeniuta et al. [2017], Yang et al. [2017] constrain the receptive field of the decoder, limiting the window of autoregression, however, this is not readily applicable to RNN architectures given their unbounded receptive field. Our proposed adversarial method falls into this category as an extension of word dropout, which is then subsumed as a special case of adversarial word dropout (Section 4.2).

**Non-uniform dropout** Previous work has recognized the benefit of adapting the dropout rate across different architectural components, though in entirely different contexts [Kingma et al., 2015, Gal et al., 2017, Achille and Soatto, 2018]. These works introduce different approaches to regularizing the magnitude of weights or activations (groups of weights). Elements that are deemed *irrelevant* during training are then dropped. For instance, variational dropout [Kingma et al., 2015] has been used to prune weights in deep neural networks [Molchanov et al., 2017]. A key difference to our approach is that instead of a regularization term, we employ an adversary that selects elements for dropout during training based on their information content not yet learned by the model.

**Exposure bias** The method by which the autoregressive decoder is trained, known as *teacher forcing*, bears a connection to *exposure bias*, which refers to the gap between training and inference [Ranzato et al., 2015]. Namely, in language generation, a trained model generates sequences at test time without access to the ground truth history that was accessible during training. Since the model was not trained to continue its predictions, this can lead to error accumulation [Ranzato et al., 2015]. The most common approach to tackle exposure bias is to switch between conditioning on ground truth and model predictions, with the latter being preferred towards the end of training [Daumé et al., 2009, Bengio et al., 2015, Ranzato et al., 2015]. In preliminary studies, we implemented *scheduled sampling* [Bengio et al., 2015] but did not find it to outperform even uniform dropout.

## 4  Adversarial Word Dropout (AWD)

### 4.1  Theoretical Basis

As a precursor to our adversarial approach, we first derive the effect of stochastic dropout on the ELBO under the autoregressive assumption (Equation 3).

**Word dropout:** Our interpretation of posterior collapse at the end of Section 2 suggests that to 'weaken the decoder' one should *restrict the available information*, such that $z$ is no longer redundant and must capture some of the restricted information. Word dropout [Bowman et al., 2016] can be seen to do this by stochastically masking the previous word $\boldsymbol{x}_{i-1}$ with probability $d_i$, which substitutes $p_\theta(\boldsymbol{x}_i|\boldsymbol{x}_{<i}, z)$ with $p_\theta(\boldsymbol{x}_i|\boldsymbol{x}_{<i-1}, z)$ in Equation 3, i.e.:

$$\text{ELBO}_{\theta,\phi}^{\textbf{WD}}(\boldsymbol{x}) = \int_z q_\phi(z|\boldsymbol{x})\Big\{ \sum_i (1-d_i)\log p_\theta(\boldsymbol{x}_i|\boldsymbol{x}_{<i}, z) + d_i \log p_\theta(\boldsymbol{x}_i|\boldsymbol{x}_{<i-1}, z) \Big\} - \text{KL} \quad (4)$$

$$= \int_z q_\phi(z|\boldsymbol{x})\Big\{ \sum_i \log p_\theta(\boldsymbol{x}_i|\boldsymbol{x}_{<i}, z) - \sum_i d_i \underbrace{\log \frac{p_\theta(\boldsymbol{x}_i|\boldsymbol{x}_{i-1}, \boldsymbol{x}_{<i-1}, z)}{p_\theta(\boldsymbol{x}_i|\boldsymbol{x}_{<i-1}, z)}}_{\text{PMI}(\boldsymbol{x}_i, \boldsymbol{x}_{i-1}|\boldsymbol{x}_{<i-1}, z)} \Big\} - \text{KL} \quad (5)$$

Equation 5 shows the effect (in expectation) of applying word dropout when maximising the ELBO with an autoregressive decoder in Eq 3 (of which 'KL' denotes the last term). In effect, a weighted sum is introduced with weights given by dropout probabilities $d_i$ and components that define conditional *point-wise mutual information* (PMI), where classic point-wise mutual information between two variables is defined as $\text{PMI}(\boldsymbol{x}_1, \boldsymbol{x}_2) = \frac{p(\boldsymbol{x}_1|\boldsymbol{x}_2)}{p(\boldsymbol{x}_1)}$. Each PMI term in Eq 5 captures the information that one sequence element $\boldsymbol{x}_{i-1}$ has regarding the next $\boldsymbol{x}_i$, *over and above* any information from earlier elements $\boldsymbol{x}_{<i-1}$ and the latent state $z$. Since the dropout-adapted ELBO (Eq 5) is maximised

w.r.t. $\theta$ and $\phi$, the weighted PMI term is minimised, which is only achievable by *increasing the information learned by the model*: extracted from $\boldsymbol{x}_{<i-1}$ or captured in $z$. Interestingly this shows that word dropout quite literally follows the earlier intuition and 'weakens' the decoder by restricting *information*. Thus, word dropout leads to a looser variational objective, i.e. $\text{ELBO}_{\theta,\phi}^{\textbf{WD}}(\boldsymbol{x}) \leq \text{ELBO}_{\theta,\phi}^{\textbf{AR}}(\boldsymbol{x})$. Rainforth et al. [2018] has argued that tighter bounds are not necessarily better. Hence, our analysis provides a justification for using word dropout.

**The minimax objective:**   Under uniform dropout, all $d_i$ are equal over a sequence, and using dropout to 'push information into $z$' is applied evenly. However, some sequence elements may hold more information than others, e.g. in language, given syntactic rules and a summary of the previous text, some words may be highly predictable without specifically knowing which word preceded them, whereas others may depend heavily on their predecessor despite the other information. This suggests that information content may be non-uniform and so applying dropout non-uniformly may be more appropriate. We, therefore, propose *targeted* dropout of elements according to their 'incremental' information content defined by the PMI terms, i.e. that not yet learned by the model. Further, since such incremental information content is not explicitly computed and will vary over training as the model learns, we train an adaptive dropout schedule to continually maximise the information dropped, and so minimize Eq 5, with respect to $d_i$. Since Eq 5 is maximised with respect to all other parameters, this leads to a minimax objective.

We introduce the two 'players' in the proposed minimax setting: $\text{VAE}_{\phi,\theta}$, trained to maximize the ELBO, and the adversary $\text{A}_\psi$ trained to do the opposite by determining each probability $d_i$ of dropping out the previous sequence element $\boldsymbol{x}_{i-1}$, preventing it from helping to predict its successor $\boldsymbol{x}_i$. The full objective for an observed sequence is given by

$$\max_{\phi,\theta} \min_{\psi} \; \mathbb{E}_{\boldsymbol{x}\sim p_*(\boldsymbol{x})}\left[\mathcal{L}_{\psi,\phi,\theta}(\boldsymbol{x}) + \mathcal{R}_\psi(\boldsymbol{x})\right] \tag{6}$$

where $\mathcal{R}_\psi(\boldsymbol{x})$ is a regularization term explained below (Section 4.3) and

$$\mathcal{L}_{\psi,\phi,\theta} = \mathbb{E}_{q_\phi(z|\boldsymbol{x})}\left[\sum_i \log p_\theta(\boldsymbol{x}_i|\text{mask}_{K,\psi}(\boldsymbol{x}_{i-1};\boldsymbol{x}), \boldsymbol{x}_{<i-1}, z)\right] - \text{KL}(q_\phi(z|\boldsymbol{x})||p_0(z)) \tag{7}$$

$\mathcal{L}_{\psi,\phi,\theta}(\boldsymbol{x})$ is a modified version of the ELBO in Equation 5 with the additional parameters $\psi$ of the adversary $\text{A}_\psi$. The adversary manifests via the masking operator $\text{mask}_{K,\psi}(\cdot\,;\boldsymbol{x})$, conditioned on the entire input sequence $\boldsymbol{x}$. Since unconstrained minimisation of Eq 7 w.r.t $\psi$ would cause all elements to be dropped out, a constraint $K$ is introduced such that exactly $K$ elements of $\boldsymbol{x}$ are dropped during autoregressive decoding. In practice, $K$ is sequence length-dependent, treated as a hyperparameter that controls the level of adversarial regularization. The adversary plays no role at test time.

### 4.2   Implementation

A high-level overview of our framework is shown in Figure 1. An encoder maps the input sequence into a hidden representation to produce the parameters of $q_\phi(z|x)$, from which $z$ is sampled using the 'reparametrization trick' [Kingma and Welling, 2014, Rezende et al., 2014]. We implement the decoder using a unidirectional LSTM [Hochreiter and Schmidhuber] for consistency with the RNN-VAE network of Bowman et al. [2016], because LSTMs are the most popular backbone for sequence VAEs and to facilitate comparison with prior work. Inspired by the recent work of [Dieng et al., 2019], we choose a *Double-LSTM* recurrent unit that elucidates the *skip-connections* to promote higher latent information content flow (see Appendix A for details).

As described in Section 4.1, the main innovation of our proposed method is the *adversarial* training procedure: instead of dropping out decoder inputs uniformly at random, we introduce a trainable adversary to drop words most important for the VAE to reconstruct the original sequence. Below we describe implementation details of the adversary, specifying the relevant components in Figure 1. A more comprehensive figure specific to the implemented architecture is provided in Appendix B. Note that the theoretical framework in Section 4.1 permits many alternative approaches for generating per word dropout probabilities $d_i$, such as attention mechanisms mentioned above. We leave further refinement of the adversarial architecture to future work.

**Producing dropout scores**   To determine which elements to drop out of a sequence, the adversary $\text{A}_\psi$ samples a *score* $s_i \in \mathbb{R}$ for each sequence element $\boldsymbol{x}_i$. Sequence elements with the $K$ smallest

| | PPL ↓ | -ELBO ↓ | KL ↑ | MI ↑ | PPL ↓ | -ELBO ↓ | KL ↑ | MI ↑ |
|---|---|---|---|---|---|---|---|---|
| **Existing sequence VAEs** | | Yahoo | | | | Yelp | | |
| CNN [Yang et al., 2017] | 63.90 | - | 10.0 | - | 41.1 | - | **7.6** | - |
| Lagging [He et al., 2019] | - | **328.4** (0.2) | 5.7 (0.7) | 2.90 | - | 357.2 (0.1) | 3.8 (0.2) | 2.4 |
| SA [Kim et al., 2018] | 60.40 | - | 7.19 | - | - | - | - | - |
| Skip [Dieng et al., 2019] | 60.90 | 330.3 | **15.05** | 7.47 | - | - | - | - |
| FBP [Li et al., 2019] | 59.51 | 330.3 | 15.02 | - | - | - | - | - |
| **Our sequence VAE** | | | | | | | | |
| unregularized | 60.30 | 328.8 (0.2) | 4.2 (0.2) | 3.14 | 40.1 | 356.4 (0.2) | 2.3 (0.2) | 1.0 |
| + word dropout [0.4] | 59.55 | 329.5 (0.4) | 14.4 (0.4) | **13.6** | 38.5 | **354.2** (0.3) | 5.9 (0.4) | 4.9 |
| + AWD (*ours*) [0.3] | **59.05** | **328.4** (0.3) | 14.4 (0.4) | **13.6** | **38.2** | **354.2** (0.3) | **6.5** (0.4) | **5.8** |

Table 1: **Results of text modeling on the Yahoo and Yelp datasets.** Standard deviations are provided in the brackets. Squared bracket contains the dropout rate $DR$. PPL – perplexity; ELBO – evidence lower bound; KL - in Eq (7); MI – mutual information $I(\boldsymbol{X}; Z)$ from Section 2.

scores are dropped. The scores are sampled from distributions $p_\psi(s_i|\boldsymbol{x})$ conditioned on the input sequence $\boldsymbol{x}$, modelled as a series of Gaussians with mean ($\mu$) and variance ($\sigma$) parameterised by the outputs of a unidirectional LSTM. Scores are generated using the reparameterization trick, reducing gradient variance [Kingma and Welling, 2014]. Thus, $p_\psi(s_i|\boldsymbol{x})$ is described as

$$s_i|\boldsymbol{x} \sim \mathcal{N}(\mu_i, \sigma_i; \boldsymbol{x}, \psi) \qquad \text{where,} \quad [\mu_i, \sigma_i] = \text{Linear}_\psi(\text{LSTM}_{i,\psi}(\boldsymbol{x}))$$

The stochastic generation of scores gives the adversary an 'exploratory' capability during training, e.g. preventing it becoming 'stuck' on a set of sequence elements with high information content that the model cannot learn, meaning that their PMI terms (in Eq 5) remain large and the adversary repeatedly selects them to minimise the ELBO.

**Top-K word selection** Based on the sampled scores $\boldsymbol{s} = \{s_i\}_{i=1}^T$, the subset of $K$ words with the smallest scores are masked during decoding. To estimate gradients of the objective with respect to the parameters $\psi$, we use a stochastic softmax trick from Paulus et al. [2020a]: during training, the stochastic subset selected in the forward pass is relaxed to admit a (biased) reparameterization gradient in the backward pass. In our experiments, we use a straight-through variant [Bengio et al., 2013, Paulus et al., 2020b] of the trick for our method to resemble word dropout, i.e. to produce discrete values $\in \{0, 1\}$. The relaxation is only used in the backward pass to compute the gradient estimator.

**Gradient reversal** The final component of the adversarial network is the *gradient reversal* layer [Ganin et al., 2016]. In the forward pass, the layer performs no transformation to the input; in the backward pass, the gradients are negated. Gradient reversal offers a computationally simple method to ensure that the parameters $\psi$ of $A_\psi$ are updated such that ELBO is minimized. If $\boldsymbol{I}$ is an identity matrix, the 'pseudo-function' of gradient reversal can be described as

$$f(\boldsymbol{x}) = \boldsymbol{x} \quad \text{(forward pass)} \qquad \frac{\partial f(\boldsymbol{x})}{\partial \boldsymbol{x}} = -\boldsymbol{I} \quad \text{(backward pass)}$$

### 4.3 Optimization challenges

Since our adversarial dropout network is fully differentiable, it is readily optimized by gradient methods, such as backpropagation. However, the magnitudes of the word dropout scores $\boldsymbol{s}$ sampled from distributions parameterized by an LSTM are unconstrained. Controlling the magnitude of the scores was found to be important for the adversary to maintain its exploratory capability (discussed above). We therefore add a KL-divergence between the distribution of each score $p(s_i|\boldsymbol{x}_i)$ and a standard Gaussian $p_0(s_i) = \mathcal{N}(s_i; 0, 1)$ to 'regularize' scores in Eq (6), subject to a scalar $\lambda > 0$:

$$\mathcal{R}_\psi(\boldsymbol{x}) = \lambda \sum_i \text{KL}(p_\psi(s_i|\boldsymbol{x}) \,||\, p_0(s_i)) \tag{8}$$

**Adversarial vs. random word dropout**   Standard word dropout [Iyyer et al., 2015, Bowman et al., 2016] can be seen as a special case of adversarial word dropout. By setting $\lambda$ sufficiently high, the regularization term in Eq 8 can be made to dominate such that $p_\psi(s_i|\boldsymbol{x}) \approx p_0(s_i)$ and all scores $s_i$ are effectively sampled from a standard Gaussian $p_0(\boldsymbol{s}_i)$ and words are dropped approximately uniformly. On the other hand, for small values of $\lambda$, $A_\psi$ will learn which words $\text{VAE}_{\phi,\theta}$ relies upon most to accurately decode the sequence and target those for drop out. As a result, whilst adding another hyperparameter to calibrate, $\lambda$ offers a simple way to moderate the difference between adversarial word dropout and standard uniform word dropout, which we consider in our experiments.

## 5   Experiments

Here we present the results of our experiments that: **(i)** demonstrate that a sequence VAE trained with adversarial word dropout (AWD) outperforms other sequence VAEs; it achieves improved sentence modeling performance and/or improved informativeness of the latent space; **(ii)** examine the contributions and behaviour of the adversarial network's components and hyperparameters, and **(iii)** qualitatively study the trained adversary and VAE.

**Datasets**   We conducted experiments on 4 different datasets: *Yahoo* questions and answers [Yang et al., 2017], *Yelp* reviews [Yang et al., 2017], Penn Tree Bank (PTB) Marcus et al. [1993] and downsampled Stanford Natural Language Inference (*SNLI*) corpus [Bowman et al., 2015, Li et al., 2019]. Yahoo, Yelp and PTB datasets were used in many previous works [Yang et al., 2017, Kim et al., 2018, He et al., 2019, Fu et al., 2019, Dieng et al., 2019] hence were used to benchmark our proposed method against comparable related works and also against standard word dropout. Yahoo, Yelp and PTB contain sentences with average lengths of 78, 96 and 22 words respectively, while SNLI sentences are much shorter with an average length of 9 words and are more suitable for qualitative studies. Yahoo, Yelp and SNLI datasets contain 100K sentences in the training set, 10K in the validation set, and 10K in the test set, while PTB is much smaller with a total of 42K sentences.

**Experimental setup**   We (re-)implement the standard VAE, a VAE with standard uniform word dropout and a VAE with our adversarial dropout method. On each dataset, we performed the same grid search over both learning rate (from $\{0.0001, 0.001, 0.1, 1\}$) and dropout rate R (from $\{0.2, 0.3, 0.4, 0.5\}$) for both the word dropout baseline and our method. This gives 16 different hyperparameter configurations for each method on each dataset. For training, we also use an exponential learning decay of 0.96 as in [Li and Arora, 2019], increased the hidden state sze of the decoder LSTM from 1024 to 2048 (except on SNLI), applied Polyak averaging [Polyak and Juditsky, 1992] with a coefficient of 0.9995 and used *KL annealing* [Bowman et al., 2016].

We apply early stopping based on validation ELBO and repeat each experiment for five different random seeds to report standard deviations. All experiments are performed on a 12GB Nvidia TitanXP GPU with an average run time of 4 hours for Yelp and Yahoo and 1 hour for SNLI.

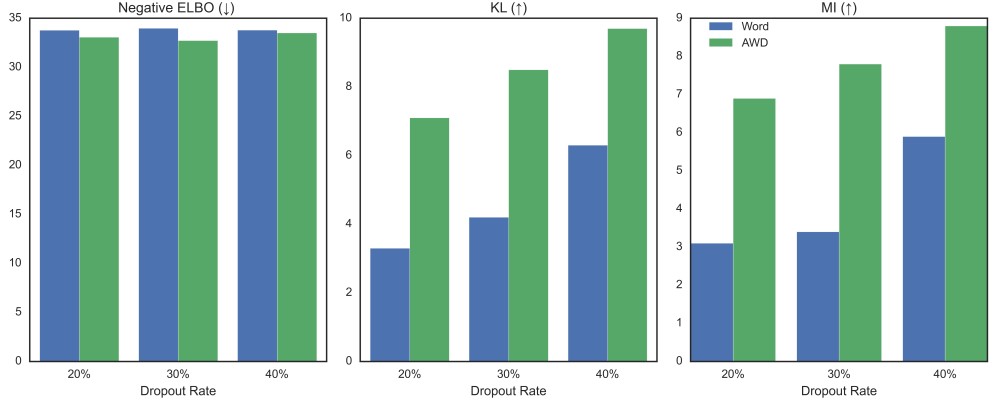

Figure 2: Adversarial vs uniform word dropout (SNLI), with ablation of the various dropout rates.

**Metrics**   Overall model performance is assessed based on two main aspects: **(i)** *sentence modeling* – measured with respect to ELBO and perplexity (PPL). ELBO and PPL quantify the ability of the trained decoder to recognise or generate natural language, reflecting the quality of density estimation; **(ii)** *latent space informativeness* – measured with respect to the KL term in Eq (7) and mutual information $I(\boldsymbol{X}; Z)$ between observed and latent variables. $I(\boldsymbol{X}; Z)$ is computed using the procedure of Hoffman and Johnson [2016], as implemented by Dieng et al. [2019]. As in comparable works [Yang et al., 2017, Kim et al., 2018, He et al., 2019, Dieng et al., 2019, Li et al., 2019], hyperparameters are set based on a balanced assessment of these metrics.

**Adversary hyperparameters**   To apply the global dropout rate $R$ to the number $K$ of elements to be dropped out we use $K = round(R \times T)$ where *round* computes the closest integer. The additional hyperparameter $\lambda$ in Eq 8 was set globally for all datasets to $\lambda = 1$. This was based on an initial exploratory analysis on the Yahoo dataset, where we grid searched $\lambda$ in $\{0.001, 0.01, 0.1, 1, 5, 10\}$ for various combinations of dropout and learning rates to find that $\lambda = 1$ consistently achieved best validation performance. Please refer to the detailed analysis of $\lambda$ in the appendix.

## 5.1   Quantitative analysis

Table 1 compares **(i)** the VAE trained with no dropout; **(ii)** the VAE trained with uniform word dropout [Bowman et al., 2016] (dropout rate $R = 0.4$ found to be best); **(iii)** a VAE trained with adversarial word dropout ($R = 0.3$ found best); and **(iv)** previously reported results for comparable models (Section 3). Our adversarial dropout method trains models that achieve better sentence modeling (lower ELBO, PPL) with an equally informative latent space (Yahoo, PTB) or more informative latent space (higher KL, MI) while maintaining sentence modeling performance (Yelp) and improves both metrics on SNLI (Appendix Table 5) . Thus, it allows users to more effectively trade-off sentence modelling and informativenes of the latent space than standard word dropout. The gains are modest in size, but larger on SNLI and PTB, and comparable to those improvements reported in previous word, e.g., [He et al., 2019] (Appendix Table 4, 5). Our method also compares favourably to previous models, consistently achieving an improved balance between sentence modeling performance (e.g. PPL) and an informative latent space (e.g. MI). We note also that the vanilla VAE does not obtain a lower perplexity than either VAE with word dropout, as might be anticipated if dropout were an arbitrary 'regularisation' method that may improve the latent space but at a cost to sequence modeling performance. This supports our theoretical analysis indicating that word dropout is not a typical 'regularizer' (Section 4.1), rather that it disrupts the flow of available information during training of the autoregressive decoder, forcing it to compensate by storing information in the latent space. Figure 2 compares adversarial and uniform word dropout, varying the dropout rate on the SNLI dataset. For any given dropout rate, adversarial dropout learns a more informative latent space (with higher KL and MI) metrics with comparable to lower negative ELBO.

**The role of $\lambda$**   The hyperparameter $\lambda$ can be seen to determine the *exploration-exploitation trade-off* of the adversary. As shown in Figure 3, for small values of $\lambda$, the magnitudes and standard deviations of word dropout scores grow large, causing the distribution of dropped words to become concentrated. For large values of $\lambda$, scores become very small with low variance and the adversary converges towards uniform word dropout. The right part of Figure 3 depicts why $\lambda = 1$ is a good choice.

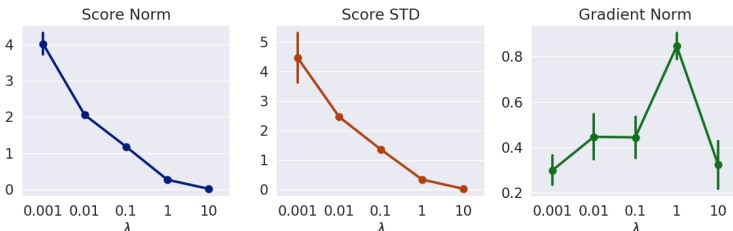

Figure 3: The role of $\lambda$. On Yahoo dataset, we computed various statistics from the data collected across iterations during one training run; *(left and mid)* Mean $\ell_1$-norm and standard deviation of dropout score vectors ($\boldsymbol{s}$); *(right)* Gradient norm of the adversary – signifies the magnitude of the parameter updates and hence the quality of learning.

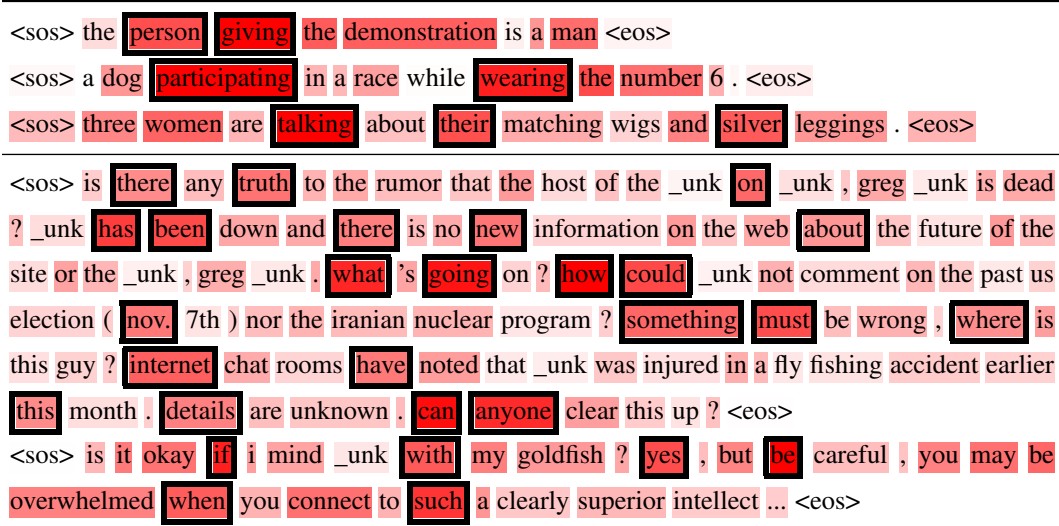

Table 2: Analysis of the Adversary. For selected SNLI *(upper)* and Yahoo *(lower)* sentences, word dropout scores are from a trained adversary and normalized per sentence. Darker colouring indicates a higher dropout probability. Boxed words are those selected to be dropped.

<sos> = start of the sequence token; <eos> = end of the sequence token; _unk = unknown token.

**Qualitative analysis** We obtain further insight into what the adversary learns by analyzing the word dropout scores for different sentences. Table 2 shows that the adversary applies lower dropout probabilities to less informative words such as 'unknown' tokens that replace all out-of-dictionary words, and so offer little information about the next word. Depending on the data semantics, the adversary selects different types of words: for SNLI, verbs tend to be picked that explain the activity, e.g. *working* and *participating*; for Yahoo, words are identified that carry question semantics, e.g. *what*, *how*, *if*, and *when*. Figure 4 (a) shows a quantitative analysis of dropout saliency map across different part-of-speech (POS) tags. Verb (verb), interjections (intj), and nouns (noun) have higher saliency scores (higher chances of being dropped) compared to punctuation (punc), determiners (det), and the start (sos) and end tokens (eos) which are relatively easier to predict given previous words. We also show that adversarial training learns a useful generative model with meaningful latent space by interpolating between sentences (Table 3). Computing BERT $F_1$ score Zhang et al. [2020] between the interpolated sentences with the source and target sentence shows the increasing trend toward the target sentence for each interpolation (as each interpolated sentence is getting away from the source and closer to the target) and decreasing for the source sentence (Figure 4 (b)).

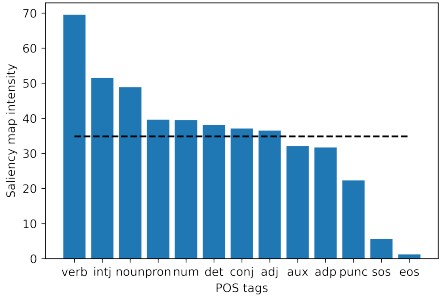
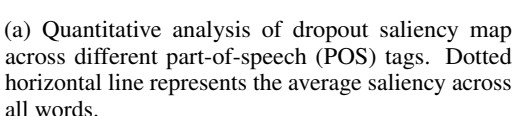
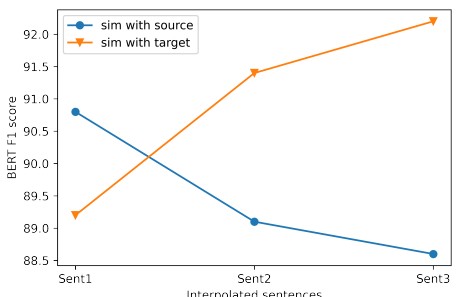

(a) Quantitative analysis of dropout saliency map across different part-of-speech (POS) tags. Dotted horizontal line represents the average saliency across all words.

(b) BERT $F_1$ score computed between the interpolated sentences with the source and target sentence. *Sent* represents the interpolated sentences (set to three in our analysis).

Figure 4: Quantitative analysis of the interpolations and saliency map presented in the paper.

# 6 Conclusions

In this work, we address the phenomenon of *posterior collapse* that occurs in autoregressive variational autoencoders (VAEs) used for sequential data. To mitigate posterior collapse, we propose a novel adversarial approach that *learns to drop words based on their information content*, leading to an improvement in both the learned latent structure and sequence modeling performance. We theoretically derive the effect of word dropout to show that, during training, it removes *incremental* information that each word provides about the next above that available from earlier words or the latent variable $z$. The only way for the model to compensate and minimise the incremental information lost is to *learn more information in the latent space*. We believe this finding provides novel and interesting insight as a means of *manipulating information* within a hierarchical latent variable model since dropout is used to 'push' specific information into the latent variable. For future work, it may be interesting to extend our method to transformer-based architectures [Vaswani et al., 2017] that can exacerbate posterior collapse when used for decoding and self-supervised learning, where input (words in NLP, image patches in computer vision) are often masked randomly [Devlin et al., 2019, He et al., 2022], but a learnt policy might improve performance or convergence.

Sequence VAEs are a promising framework that, beyond purely autoregressive models [Brown et al., 2020], hold the prospect of controlled sequence generation. Improvements to such general methods may inevitably be used for good or ill, from the generation of targeted fake news on the one hand to the possibility of personalised human-computer interactions in, say, the medical domain on the other (e.g. for modeling sleep [Miladinović et al., 2019b, Nowak et al., 2021]). In future work, we plan to explore alternative implementation options, particularly of the adversary, and extend adversarial dropout to other domains, such as images, speech, or dynamical systems [Bauer et al., 2017].

---

**I 'm not sure what all the hype is about . i 've been here a few times and it 's just ok . nothing special . I would n't go out of my way to come here .**

*I 've been here a few times and it 's always been good . the food is good , but the service is not so great .*

*I 've been here a few times and it 's always been good . i 've had the chicken and waffles and the service was good .*

*Great place to go for a quick bite to eat . the food is great and the service is great . i have been here a few times and have never been disappointed .*

**Great food , great service , and great service . i 've been here a few times and have never been disappointed**

---

Table 3: Sentence interpolation (Yelp dataset). Representations of two sentences *(top, bottom)* are obtained by feeding them through an adversarially trained VAE encoder. Three linearly interpolated representations are passed to the VAE decoder and sentences generated by greedy sampling *(middle)*.

# 7 Acknowledgments

We thank Taylor Berg-Kirkpatrick for his thoughtful insights and valuable feedback. Carl is gratefully supported by an ETH AI Centre Postdoctoral Fellowship, a responsible AI grant by the Haslerstiftung; Swiss National Science Foundation (project # 201009), and an ETH Grant (ETH-19 21-1).

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
