# Appendix

## A   Double-LSTM

Inspired by the recent work [Dieng et al., 2019] that eluci-
dates how *skip-connections* promote higher latent information
content, we introduce a simple-to-implement modification of
a standard LSTM [Hochreiter and Schmidhuber]. Double-
LSTM aims to promote the utilization of the latent variable $z$,
whilst increasing the expressive power of the autoregressive
decoder. Double-LSTM consists of two LSTM units [Hochre-
iter and Schmidhuber] as depicted in Figure 5. The first LSTM
unit is updated based on the latent variable $z$ and the previous
hidden state $h$. The second LSTM unit is updated based on
$z$, $h$ and the input word embedding $w$, which is subject to
teacher forcing and dropout. The benefit of Double-LSTM is

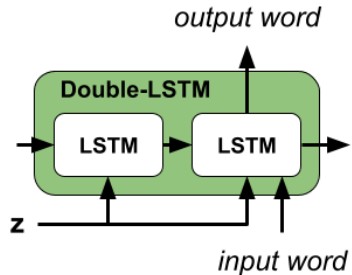

Figure 5: **Double-LSTM.**

that it provides a two-branched skip connection to link $z$ with the output word. The state update
performed by the first LSTM is guaranteed to extract information from latent states and not from
ground truth input. In practice, Double-LSTM leads to performance improvements with little cost in
terms of memory and computation time.

## B   Implementation Architecture

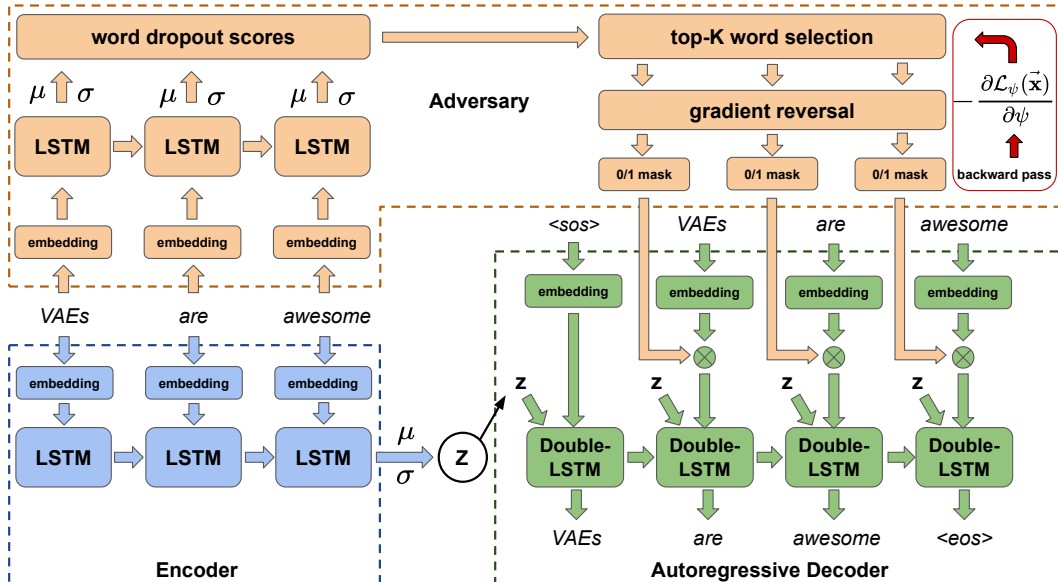

Figure 6: **Architectural details of our proposed method:** Our variant of RNN-VAE [Bowman
et al., 2016] consists of an LSTM-based sequence encoder (depicted in blue) and a Double-LSTM-
based autoregressive sequence decoder (depicted in green). During MLE training, the ground truth
sequence history is *teacher-forced* to the autoregressive decoder which reduces the utilization of
$z$. To regularize autoregressive decoding, the *adversary* $A_\psi$ (depicted in orange) obstracts VAE
by *learning to drop out* words (sequence elements) that VAE requires. Concretely, $A_\psi$ selectively
masks different words at decoder's input, effectively performing input-conditioned word dropout.
In the first stage, $A_\psi$ stochastically produces unnormalized *word dropout scores* for each word in
a sequence, using another LSTM that architecturally mirrors the one in the encoder. In the second
stage, $A_\psi$ selects a subset of $K$ 'dropouts' using the differentiable *top-K word selection* module.
Finally, to ensure that $A_\psi$ optimizes an objective that is an inverted version of the VAE objective,
*gradient reversal* [Ganin et al., 2016] is applied to negate the gradients in the backward pass.

## C   Additional Quantitative Analysis

|  | -ELBO ↓ | KL ↑ | MI ↑ |
|---|---|---|---|
| **Existing Sequence VAEs** | | | |
| Annealing Bowman et al. [2015] | 101.2 | 0.00 | - |
| $\beta$ - VAE Higgins et al. [2017] | 104.5 | 7.50 | 3.1 |
| SA Kim et al. [2018] | 102.6 | 1.23 | 0.7 |
| Cyclical Fu et al. [2019] | 103.1 | 3.48 | 1.8 |
| **Our Sequence VAE** | | | |
| unregularised | 102.6 (0.3) | 1.1 (0.1) | 0.8 (0.4) |
| + Word Dropout [0.4] | 101.4 (0.3) | 5.1 (0.2) | 4.2 (0.3) |
| + AWD [0.2] | **99.7** (0.2) | 5.1 (0.1) | **4.3** (0.3) |

Table 4: **Results of text modeling on the PTB dataset** Marcus et al. [1993]. Standard deviations are provided in the brackets. Squared bracket contains the dropout rate $DR$. ELBO – evidence lower bound; KL - KL divergence; MI – mutual information.

|  | -ELBO ↓ | KL ↑ | MI ↑ |
|---|---|---|---|
| **Existing Sequence VAEs** | | | |
| Annealing Bowman et al. [2015] | 33.08 | 1.42 | - |
| Lagging He et al. [2019] | 32.95 | 1.42 | - |
| Cyclical Fu et al. [2019] | 34.32 | 3.63 | - |
| FBP Li et al. [2019] | 34.25 | **8.99** | - |
| **Our Sequence VAE** | | | |
| unregularised | 34.61 (0.1) | 0.06 (0.04) | 0.8 (0.1) |
| + Word Dropout [0.4] | 33.82 (0.2) | 6.04 (0.6) | 5.88 (0.6) |
| + AWD [0.3] | **32.66** (0.2) | 8.01 (0.7) | **7.22** (0.8) |

Table 5: **Results of text modeling on the SNLI dataset**. Standard deviations are provided in the brackets. Squared bracket contains the dropout rate $DR$. ELBO – evidence lower bound; KL - KL divergence; MI – mutual information.

## D   Definition of "posterior collapse"

We note that that the phenomena we address is often referred to as *posterior collapse*, which could be misinterpreted as meaning that the posterior in fact *collapses* to a single point (as in an MLE or MAP estimate), particularly since for VAEs, latent posteriors are typically more concentrated than the prior. As such, it might be less ambiguous to refer to the phenomenon as *posterior dispersion* or *posterior ignorance* or similar to better capture the fact that the posterior becomes diffuse and carries no information with respect to the data.

## E   Additional Qualitative Analysis

This section provides additional qualitative experiments performed using the sequence VAE trained with the proposed adversarial training.

**I 'm not sure what all the hype is about . i 've been here a few times and it 's just ok . nothing special . I would n't go out of my way to come here .**

*I 've been here a few times and it 's always been good . the food is good , but the service is not so great .*

*I 've been here a few times and it 's always been good . i 've had the chicken and waffles and the service was good .*

*Great place to go for a quick bite to eat . the food is great and the service is great . i have been here a few times and have never been disappointed .*

**Great food , great service , and great service . i 've been here a few times and have never been disappointed**

Table 6: **Sentence interpolation on the Yelp dataset.**

---

<sos> do you think that you should be with someone you love ? if you do n't know what you are , then you should be happy . <eos>

i am not sure if he is going to get a job . i am not ready to get married but i am not sure how to get him to pay for it . <eos>

<sos> please help me with my homework ? i am a _unk student and i need to know what the average salary of a school is in the us . i am looking for a website that has a list of the average salaries of students in the united states. <eos>

<sos> what are some good websites to get free stuff on the net ? <eos>

<sos> how do i become a better person ? i 'm shy , but i do n't know how to approach a guy . what should i do ? you should be able to be friends with someone who is not interested in you . if you are shy. <sos> if you have a quadratic equation , what is the value of x ? x = -1 <eos>

<sos> how to get a _unk visa ? i have a degree in psychology and i want to know what is the process of getting a job in the us . if you are a _unk , you can apply for a job . <eos>

<sos> the u.s. has a nuclear power to stop the war in iraq ? what is the reason for the war ? the war is over . <eos>

<sos> question about jesus ? what is the name of the church that jesus is in the bible ? what is the name of the church ? <eos>

<sos> do you think it is bad for you to have a cold sore ? i have a cold sore and i have a bad breath . why do i have to pee ? it 's because it 's not a bad thing . <eos>

<sos> is there any _unk in the world ? yes , but there is no such thing as a soul mate . <eos>

Table 7: **Unconditional sentence generation based on the Yahoo dataset.**

---

**<sos> since the nfl has gone to the eight division format ? have three teams from teh same division made the playoffs in the same year not yet but could happen this year with the cowboys , giants , and redskins in the nfc east , or steelers , bengals , and ravens in the afc north .**

<sos> the dog jumps into the air to catch a toy in its mouth . <eos>

<sos> a young woman in a white shirt and black pants is playing with a young boy in a blue shirt . <eos>

<sos> the person is flying a plane . <eos>

<sos> the dogs have their owners in the air in front of a crowd of onlookers . <eos>

 <sos> the people are participating in an operation . <eos>

<sos> a woman with black hair is standing in a puddle . <eos>

<sos> a young woman is riding a bike in front of a group of people in a red dress .

<sos> people are holding up their signs in their hands . <eos>

Table 8: **Neighborhood exploration based on the Yahoo dataset.** The original sentence taken from the Yahoo dataset *(on top)* was used to infer the parameters of the posterior. We then sampled from the posterior and decoded the sentences multiple times.