# OpenReview forum: "Learning to Drop Out: An Adversarial Approach to Training Sequence VAEs"
_NeurIPS.cc/2022/Conference — NeurIPS 2022 Accept_

### Official Review · Reviewer_WbUQ · 2022-07-11

**Rating:** 5
**Confidence:** 3
**Soundness:** 3 good
**Presentation:** 3 good
**Contribution:** 2 fair

**Summary:**

The authors tackle the problem known as "posterior collapse" which arises while applying variational autoencoders (VAEs) to sequential data in an autoregressive manner. The authors extend a previous method known as "word dropout" which uniformly drops words during the autoregressive decoding process. The authors suggest to formulate the problem as a minimax optimization task. They present an adversary, that predicts the most important words (in term of information gain) and mask them. To regularize this training scheme, the authors suggest to mask only the top K words, where K is an hyperparameter of the method. Another regularization which adds exploratory factor to the process is that the adversary estimates a Gaussian distribution parameters as scores for the masking process, then the scores are sampled from this distribution. Another regularization added to the method is that the estimated adversary Gaussian distribution score for each of the words  is also being regularized using KL-divergence term to the normal Gaussian distribution.

**Questions:**

- The paper suggests the section 4.1 permits alternative approaches for generating per word dropout probabilities. It would have been helpful to discuss approaches that the authors tried and did not work for them.

**Limitations:**

I believe there is no potential negative societal impact to this work.

**Strengths And Weaknesses:**

**Strengths**
- The paper is written in a clear and easy to follow manner, high clarity.
- The approach is novel and original, although extended a previous uniform sampling method.
- The work provides some new insights to the "posterior collapse" in terms of theoretical analysis.
- The work present experimentation study which support the claims and proposed method in this paper

**Weaknesses**
- The significance of the work is not completely clear to me, I am not confident that the improvements over the baseline are significant.
- The ablation study is missing a more in-depth analysis of the parameter \lambda. It is not clear why the authors chose \lambda=1 as their choice.
- In Table 1, we do not see improvement in KL of MI over uniform sampling on the Yahoo dataset, the paper does not discuss it.

---

> ### Author Response · Authors · 2022-08-02
> **ELBO and KL/MI are two conflicting objectives, our method facilitates a better trade-off.**
>
> > “[why no] improvement in KL/ MI over uniform sampling on the Yahoo dataset[?]”
>
> Thank you for your question. As noted above, using dropout for learning sequence VAEs typically presents a trade-off between sentence modeling performance (ELBO, PPL) and latent space informativeness (KL, MI). Using our method, users can more effectively trade-off these two objectives. It is possible to achieve better sentence modeling with an equally informative latent space (e.g., Yahoo) or more informative latent space (KL, MI) while maintaining sentence modeling performance (e.g., Yelp) or both (e.g., SNLI). We investigated the particular case of the Yahoo dataset and found that increasing the global dropout rate to 40% increased mutual information to 15.4. while worsening ELBO slightly.

---

> ### Author Response · Authors · 2022-08-02
> **Here is some additional analysis on the regularization parameter $\lambda$**
>
> > “in-depth analysis of the parameter $\lambda$; why [..] choose $\lambda=1$?”
>
> Thank you for your interest in the details of our method, in particular, the parameter $\lambda$ and the effectiveness of regularizing the adversary’s dropout distribution.
>
> Higher values of $\lambda$ result in output distributions that are more like the standard Gaussian and thus give rise to dropout policies that are more like the uniform (word dropout) baseline. Naturally, a stronger regularization will also tend to reduce the absolute magnitude of the sampled scores $s$. This is demonstrated in Figure 3 of the original draft: We trained multiple sequence VAEs at different values for $\lambda$, but otherwise identical hyperparameters on the Yahoo dataset, and recorded the average absolute magnitude of the scores during training. In practice, limiting the absolute size of the scores $s$ is important because gradients of the differentiable top-k word selection can saturate in large values of $s$ (induced by too small $\lambda$ values). This is demonstrated in the figure linked [here](https://ibb.co/2WmYbnw). Larger values of $\lambda$ and accordingly smaller scores tend to increase the average absolute magnitude of the gradients of the adversary’s parameters along the training trajectory. This facilitates learning until the regularization becomes too strong ($\lambda = 10$).
> The regularization parameter $\lambda$ was set globally to the value of $1$ for all datasets after initial exploration of the Yahoo dataset showed the best validation performance for this choice across various dropout and learning rate configurations.
>
> In response to your concern,
> - we updated Figure 3 and the corresponding text.

---

> ### Author Response · Authors · 2022-08-02
> **Improvements are modest in size, consistent, comparable and significant.**
>
> > “the improvement is relatively marginal compared to [...] word dropout, [limited improvement]”
>
> Thank you for raising this point. While we agree that the improvements are modest in size, they are consistent and we believe they are notable.
>
> Using dropout for learning sequence VAEs typically presents a **trade-off between sentence modeling performance (ELBO, PPL) and latent space informativeness (KL, MI)**: As dropout increases, latent space informativeness tends to increase (Figure 2), as the model is forced to communicate more information via the latent sentence representation. But for high values of dropout, sentence modeling performance often deteriorates, because of the lack of teacher forcing. Our adversarial dropout method allows users to **more effectively trade-off these two conflicting objectives than standard word dropout**. It trains models that achieve better sentence modeling with an equally informative latent space (e.g., Yahoo) or more informative latent space (KL, MI) while maintaining sentence modeling performance (e.g., Yelp). The gains extend to 0.9, 1.1, 3.1 nats on Yelp, Yahoo and SNLI (newly included, see response about SNLI results) respectively. While this is modest in size, it is **comparable to other important improvements** in VAE training, e.g. IWAE (Burda et al., 2015) reports gains ranging from one to three nats, for sequence VAEs He et al. (2019, ICLR) report improvements in ELBO on Yelp of upto 1.2 nats and don’t improve ELBO on Yahoo, while increasing KL. Importantly, we repeat all of our experiments with five different random seeds to find **statistical significance** at the 5% level.
>
> In addition, we show improvements over competitor sequence VAEs. We improve ELBO by three nats over He et al. (2019) on Yelp and by nearly two nats over Li et al. (2019) and Dient et al. (2019) on Yahoo. Finally, we decided to include the PTB dataset  [Markus et al., 1993]  as a fourth dataset to test our method (see response to Lidz on SNLI). We find improvements of 1.8 nats from using adversarial dropout instead of standard word dropout. [PTB Table link](https://ibb.co/60sQ6QB)
>
>
> In response to your concern,
> - we now more clearly communicate our experimental results in Section 5
> - and we have included the additional results in the draft (main body or supplement)
>
> *This concern was shared by at least one other reviewer, we double-post the reply here to ease correspondence.*

---

> ### Author Response · Authors · 2022-08-02
> **Thank you! We made revisions based on your feedback.**
>
> **Thank you very much for your thoughtful review.** Thank you for acknowledging the clarity of the writeup, the novelty and the originality of the draft, the introduction of the new theoretical insights, and also that the experimentation study supports our hypothesis. **We have now improved the draft based on your feedback** and addressed your concerns in more detail in the comments below.

---

### Official Review · Reviewer_KBue · 2022-07-11

**Rating:** 5
**Confidence:** 3
**Soundness:** 2 fair
**Presentation:** 3 good
**Contribution:** 3 good

**Summary:**

The paper aims to investigate the problem of posterior collapse that variational autoencoders often suffer from. To tackle this issue, the authors propose an adversarial training strategy. To be specific, an adversary is employed to learn the probabilities of the decoder input dropout. The method is first demonstrated theoretically to remove pointwise mutual information and then evidenced by positive experimental results on Yahoo, Yelp, and SNLI against several baselines with double LSTMs as the decoder. A quantitative analysis follows to demonstrate the method further (e.g., the trade-off between exploration and exploitation).

**Questions:**

1). Is double lstm a part of the contributions? This misleads the point of this work.

2). Since the idea itself is fairly simple, are these notations and formulas necessary for its presentation?

3). The ablation study regarding the double lstm is necessary. In other words, experiments against various types of the decoder are required.

4). (line 152) The problems described lack the support of previous works.

5). (line 185) A wrong reference to the appendix.

6). (figure 2) What does the y-axis correspond to?

7). Tables 2 & 3 can be placed in the appendix to make some space for more experiments and studies.


**Limitations:**

Limitations are mentioned in Section 6, while it seems to be not sufficient.

**Strengths And Weaknesses:**

Strengths:1). The authors touch on the issue shared by most autoregressive decoders, that is, posterior collapse; thus, works in this direction, I believe, will benefit the whole community due to the popularity of the autoregressive decoders. 2). The method is well-motivated by the problem of posterior collapse and well-defined theoretically. The idea is fairly simple for an easy understanding and application. 3). The introduced method is successfully examined to outperform baselines by thoughtful experiments on several benchmarks. A quantitative analysis is conducted, showing specific examples of the adversary.

Weaknesses:1). Technically speaking, the contribution of this work is incremental, and its technique deep is shallow. The proposed probabilistic word dropout is not that impressive or novel. To me, it sounds like a probabilistic teacher forcing. The massive notations and formulas appear to be not that necessary for me to understand the idea. 2). The improvements on three tasks over the previous works and self-implemented baselines are marginal. Further analysis beyond the main experiments is not sufficient. 3). The backbone is constrained to be the double LSTM, while popular transformers is not involved. Although this seems to work for different encoders, pre-trained language models are not covered as well. The application of this method to more extensive model structures remains a potential concern.

---

> ### Author Response · Authors · 2022-08-02
> **LSTMs are the go-to choice for sequence VAEs**
>
> > “popular transformers [are] not involved”
>
> Thank you for raising this concern. We would like to clarify that our work focuses on avoiding posterior collapse in sequence VAEs. We chose to experiment with LSTMs, because they are the most popular choice for encoders and decoders in sequence VAEs and to maintain comparability to previous work in sequence VAEs [Yang et al., 2017, Kim et al., 2018, Dieng et al., 2019, He et al., 2019, Li et al., 2019]. Pre-trained language models do not typically elicit a (probabilistic) latent representation and are not trained with variational objectives meaning that posterior collapse is not a concern [Park and Lee 2021].
>
> Still, we do agree that there may be merit in combining adversarial dropout with transformer-based architectures. There are various possible directions. For example, recently masked autoencoders [He et al., 2021] achieved strong performance in self-supervised computer vision. To train these models, random patches of the input image are masked. It seems plausible that performance may be improved by learning a model to mask patches  in analogy to adversarial dropout as we proposed it for sequence VAEs. Sadly, this is outside scope of our current draft, but we hope that our work will inspire future research in these directions.
>
> In response to your concern,
> - we now more clearly highlight the reasons for choosing the LSTM backbone in Section 4.2.
> - and include a discussion on extensions to transformer-based models as possible future work in the conclusion.
>
> References
> - Park and Lee, 2021. Finetuining Pretrained Transformers into Variational Autoencoders.
> - He et al., 202. Masked Autoencoders Are Scalable Vision Learners.

---

> > ### Comment · Reviewer_KBue · 2022-08-07
> > **Response to author**
> >
> > Thanks for the clear response and the efforts to involve additional experiments and the updated version. Most of my concerns have been addressed, while I still would like to underline my concern regarding the ablation study of the encoder/decoder with attention-based structures. The theoretical findings should not be constrained by the model structure, and I doubt that LSTM is still the most widely used one for sequential VAE, which is also not the reason to exclude transformers. Overall, I have updated my score given the response and the updated version.

---

> ### Author Response · Authors · 2022-08-02
> **Improvements are modest in size, consistent, comparable and significant.**
>
> > “the improvement is relatively marginal compared to [...] word dropout, [limited improvement]”
>
> Thank you for raising this point. While we agree that the improvements are modest in size, they are consistent and we believe they are notable.
>
> Using dropout for learning sequence VAEs typically presents a **trade-off between sentence modeling performance (ELBO, PPL) and latent space informativeness (KL, MI)**: As dropout increases, latent space informativeness tends to increase (Figure 2), as the model is forced to communicate more information via the latent sentence representation. But for high values of dropout, sentence modeling performance often deteriorates, because of the lack of teacher forcing. Our adversarial dropout method allows users to **more effectively trade-off these two conflicting objectives than standard word dropout**. It trains models that achieve better sentence modeling with an equally informative latent space (e.g., Yahoo) or more informative latent space (KL, MI) while maintaining sentence modeling performance (e.g., Yelp). The gains extend to 0.9, 1.1, 3.1 nats on Yelp, Yahoo and SNLI (newly included, see response about SNLI results) respectively. While this is modest in size, it is **comparable to other important improvements** in VAE training, e.g. IWAE (Burda et al., 2015) reports gains ranging from one to three nats, for sequence VAEs He et al. (2019, ICLR) report improvements in ELBO on Yelp of upto 1.2 nats and don’t improve ELBO on Yahoo, while increasing KL. Importantly, we repeat all of our experiments with five different random seeds to find **statistical significance** at the 5% level.
>
> In addition, we show improvements over competitor sequence VAEs. We improve ELBO by three nats over He et al. (2019) on Yelp and by nearly two nats over Li et al. (2019) and Dient et al. (2019) on Yahoo. Finally, we decided to include the PTB dataset  [Markus et al., 1993]  as a fourth dataset to test our method (see response to Lidz on SNLI). We find improvements of 1.8 nats from using adversarial dropout instead of standard word dropout. [PTB Table link](https://ibb.co/60sQ6QB)
>
> In response to your concern,
> - we now more clearly communicate our experimental results in Section 5
> - and we have included the additional results in the draft (main body or supplement)
>
> *This concern was shared by at least one other reviewer, we double-post the reply here to ease correspondence.*

---

> ### Author Response · Authors · 2022-08-02
> **Reflecting on technical concerns**
>
> > “Technically speaking [...] not that impressive or novel; [too simple]”
>
> We politely disagree. We believe our approach is theoretically well grounded and the technical analysis is insightful. We show that masking words from the input sentence removes pointwise mutual information and thus leads to a looser variational objective. Neither the original draft that proposed word dropout [Iyyer et al., 2015] nor the one that popularized it [Bowman et al., 2016] noted these two findings. Further, it has been argued by [Rainforth et al., 2018] that tighter bounds are not necessarily better. Hence, our analysis provides a justification for word dropout which is to the best of our knowledge new. It also yields an interesting additional perspective on our approach: Dropping out different words leads to different variational objectives and our method learns which objective to choose using an adversarial approach.
>
> However, we do agree this message must be conveyed more clearly in our draft. Accordingly,
> - we have revised Section 4.1 and made the connection to Rainforth et al. explicit.
>
>
> References
> - Rainforth et al., 2018. Tighter variational bounds are not necessarily better.

---

> ### Author Response · Authors · 2022-08-02
> **Thank you! We made revisions based on your feedback.**
>
> **Thank you very much for your review** and for acknowledging the importance of the problem (posterior collapse), the motivation of the method and its successful experimental examination. **We have now improved the draft based on your feedback** and addressed your concerns in more detail in the comments below.

---

### Official Review · Reviewer_Lidz · 2022-07-12

**Rating:** 6
**Confidence:** 4
**Soundness:** 3 good
**Presentation:** 4 excellent
**Contribution:** 3 good

**Summary:**

This paper aims to address the posterior collapse issue that occurred during training sequence VAEs. To this end, the authors first provide theoretical support for the effectiveness of word-level Dropout regularization to mitigate this problem. Then, they propose a new non-uniform, data-dependent Dropout scheme by introducing the adversarial training strategy with proper regularization. Finally, the empirical effectiveness of the proposed method is demonstrated with three different datasets by comparing it with other sequence VAEs.



**Questions:**

Please carefully read the above cons and address the related questions and suggestions.

**Limitations:**

The authors have adequately addressed their work's limitations and potential negative social impact at the end of the conclusions.

**Strengths And Weaknesses:**

### **Pros.**

- **Clarity**. Overall, the writing is clear and easy to follow. In addition, the organization of the main draft is well-established.
- **Well motivated problem and intuitive solution**. Resolving the posterior collapse issue is an important and interesting problem to solve. Also, the provided theoretical support and the proposed method are intuitive.

### **Cons.**

- **Insufficient empirical demonstration**. Although the clear advantage of the provided theoretical support and intuitive method, there are clear limitations in the current empirical results.
  1. *Fairness with the hyper-parameter search*: as presented in lines 271-274, the authors conduct an extensive hyper-parameter search for the proposed method. However, such a hyper-parameter search is not conducted for the baselines, e.g., word Dropout with a fixed-rate 0.4. As the optimal hyper-parameter largely varies on each dataset, the authors should conduct the same hyper-parameter search procedure to baselines for a fair comparison.
  2. *Limited improvement*: even though the adversarial training scheme might increase the overall computations during the training, the improvement is relatively marginal compared to the vanilla Word Dropout.
  3. *Absence of the results with SNLI*: the authors mention that they have conducted experiments on three different datasets (lines 237-239). However, the results on only two datasets are presented in Table 1. Although they present some qualitative results on SNLI, quantitative results (like Table 1) should be provided to demonstrate its effectiveness in various scenarios.
  4. *Quantitative support in addition to qualitative results in Tables 2 and 3*: although the provided qualitative results are intuitive and helpful to demonstrate the benefit of addressing the posterior collapse issue with the proposed method, they also have some concerns for the generalization to other samples. Therefore, if the authors can provide additional quantitative results, such as alignment with the saliency map for Table 2 or the similarity between original samples and interpolated ones, then it would significantly enhance the merit of the proposed method.

### **Other Comments.**

- Minor comments
    - Mis-allocation; line 185 and line 190.

---

> ### Author Response · Authors · 2022-08-02
> **We now provide quantitative analysis to support Table 2 and 3.**
>
> > “additional quantitative results [to support] Table 2 and Table 3”
>
> Thank you for your concern. In response, we performed two additional experiments:
> - **POS tags**. We averaged the saliency values from Table 2, on a scale of 0-100, for each word across 100 sentences, and categorized them based on the part-of-speech tag. The results are linked [here](https://ibb.co/PxGCCfp). As observed qualitatively, we find that verbs top the list in terms of saliency intensity. In other words, we verify that the adversary tends to pick the words that explain the activity and carry sentence semantics.
> - **BERT F1 similarity scores** computed using "bert_score_library" from HuggingFace. For 25 interpolated sentences, we computed the similarity of interpolated sentences against the source and target sentences. The results are linked [here](https://ibb.co/D7Hpfbt). The plot shows a clear downward/upward trend when moving from source to target sentences in terms of similarity score.
>
> In response to your concern,
> - we have included the additional results in the supplementary material.

---

> ### Author Response · Authors · 2022-08-02
> **We included the quantitative results for SNLI + new results on PTB dataset**
>
> > “only two datasets [...], quantitative results [on SNLI] should be provided”
>
> We previously included some of the quantitative results of SNLI in Figure 2, but we agree there is merit in including the full results as we previously did for Yahoo and Yelp data in Table 1. The **full table for SNLI is linked [here](https://ibb.co/mNJDR8g) and the results are also included in the draft/ supplementary material**. Our method (AWD) improves over word dropout in all metrics. ELBO improves by slightly more than one nat, KL improves by nearly two nats, and mutual information improves by over two nats. AWD outperforms competitor sequence VAEs in ELBO at much higher KL, which is only matched by FBP albeit at worse sentence modeling capacity.
>
> In further response to your concern, we also ran experiments for an **additional fourth dataset, PTB** (Marcus et al, 1993). The results are linked [here](https://ibb.co/60sQ6QB) and also included in the supplementary material. Our model (AWD) improves over word dropout in ELBO by nearly two nats, while maintaining the informativeness of the latent space. AWD outperforms competitor VAEs, achieving much better ELBO. Its KL is only exceeded by Higgins, which however gives worse mutual information.
>
> - We find these additional experimental findings encouraging and have included them in the draft in response to your concern.
>
> References
> - Marcus et al., 1993. Building a Large Annotated Corpus of English: The Penn Treebank.

---

> ### Author Response · Authors · 2022-08-02
> **Improvements are modest in size, consistent, comparable and significant.**
>
> > “the improvement is relatively marginal compared to [...] word dropout, [limited improvement]”
>
> Thank you for raising this point. While we agree that the improvements are modest in size, they are consistent, comparable and significant. We do believe they are notable.
>
> Using dropout for learning sequence VAEs typically presents a **trade-off between sentence modeling performance (ELBO, PPL) and latent space informativeness (KL, MI)**: As dropout increases, latent space informativeness tends to increase (Figure 2), as the model is forced to communicate more information via the latent sentence representation. But for high dropout values, sentence modeling performance often deteriorates, because of the lack of teacher forcing. Our adversarial dropout method allows users to **more effectively trade-off these two conflicting objectives than standard word dropout**. It trains models that achieve better sentence modeling with an equally informative latent space (e.g., Yahoo) or more informative latent space (KL, MI) while maintaining sentence modeling performance (e.g., Yelp). The gains extend to 0.9, 1.1, 3.1 nats on Yelp, Yahoo and SNLI (newly included, see the response about SNLI results) respectively. While this is modest in size, it is **comparable to other important improvements** in VAE training, e.g. IWAE (Burda et al., 2015) reports gains ranging from one to three nats, for sequence VAEs He et al. (2019, ICLR) report improvements in ELBO on Yelp of up to 1.2 nats and don’t improve ELBO on Yahoo while increasing KL. Importantly, we repeat all of our experiments with five different random seeds to find **statistical significance** at the 5% level.
>
> In addition, we show improvements over competitor sequence VAEs. We improve ELBO by three nats over He et al. (2019) on Yelp and by nearly two nats over Li et al. (2019) and Dient et al. (2019) on Yahoo. Finally, we decided to include the PTB dataset  [Markus et al., 1993]  as a fourth dataset to test our method (see the response on SNLI). We find improvements of 1.8 nats from using adversarial dropout instead of standard word dropout. [PTB results link](https://ibb.co/60sQ6QB).
>
> In response to your concern,
> - we now more clearly communicate our experimental results in Section 5
> - and we have included the additional results in the draft (main body or supplement)
>
> *This concern was shared by at least one other reviewer, we double-post the reply here to ease correspondence.*

---

> ### Author Response · Authors · 2022-08-02
> **We did grid-search the same hyper-parameters for the word dropout baseline.**
>
> > “word dropout fixed [at rate 0.4 …], conduct same hyperpameter search for fair comparison.”
>
> Thank you for raising this point. We apologize for not clarifying the experimental protocol in more detail. We do agree that a fair evaluation of the (word dropout) baseline and our adversarial dropout method on the same computational budget for tuning hyperparameters is essential to draw any conclusions about the practical merit of using either method. And we did just this. We give a clear account of our experimental protocol below:
>
> On each dataset, we performed the same grid search over both learning rate (from $[0.0001, 0.001, 0.1, 1]$) and dropout rate R (from $[0.2, 0.3, 0.4, 0.5]$) for **both** the word dropout baseline and our adversarial dropout method. Each of the 16 resultant configurations was evaluated on the same five random seeds, resulting in 80 independent runs for each method on each dataset.
>
> The remaining training hyperparameters (learning rate decay, batch size etc.) were set globally to values that previous work [Bowman et al., 2015, Kim et al., 2018, He et al., 2019] found to work best. But our adversarial dropout posits the additional hyper-parameter $\lambda$ to regulate the adversarial dropout distribution for which we could not naturally draw on previous work. Therefore, we performed an initial exploratory experiment on the Yahoo dataset only, where we grid searched $\lambda$ in $[0.001, 0.01, 0.1, 1, 5, 10]$ for various combinations of dropout and learning rates to find that $\lambda=1$ consistently achieved best validation performance. Therefore, we set $\lambda$ globally to 1 in all experiments and recommend this as the default. Note that $\lambda$ is likely to generalize, because the scorewise regularization is independent of the input sequence length and the stochastic softmax trick we used scales gracefully to higher dimensions (see more analysis on $\lambda$ in our response to reviewer WbUQ) .
>
> In response to your concern,
> - we have updated Section 5 in the draft to clarify the experimental protocol.

---

> ### Author Response · Authors · 2022-08-02
> **Thank you! We made revisions based on your feedback.**
>
> **Thank you very much for your thoughtful review**. Thank you for acknowledging the clarity, the theoretical support, the significance of the problem, the intuitiveness of the solution, and also the technical soundness of our draft. We have taken your concerns into account and have **improved the draft based on your feedback**. Please see detailed responses below.

---

> ### Comment · Reviewer_Lidz · 2022-08-07
> **After rebuttal**
>
> Thank you very much for the response. I appreciate the effort that the authors put into addressing my questions. I believe that the above results and discussion can significantly improve the quality of the manuscript. My major concerns are mostly addressed, but I have one more question. In the response with "Improvements are modest in size, consistent, comparable and significant.", the authors argued that AWD allows users to more effectively trade-off these two conflicting objectives (KL and MI vs. ELBO and PPL) than standard word dropout. But, only the results of KL and MI with different dropout rates are now presented; hence it's not enough to support this claim. Is there any experimental support for ELBO and PPL with varying dropout rates? Namely, as the KL and MI have similar meanings, it would be natural to substitute one of the bar figures in Figure 2 with the one with ELBO or PPL.

---

> > ### Author Response · Authors · 2022-08-08
> > **Spot on! We revised Figure 2.**
> >
> > You are absolutely right!
> >
> > **We revised Figure 2 to include both test ELBO and PPL** and you can find it [here](https://ibb.co/LN2BrKv). It now shows all four metrics for both the word dropout baseline and our adversarial word dropout (AWD) at varying dropout rates on test SNLI. As expected, both KL and MI (higher is better!) tend to increase with more dropout for both methods. But AWD uses dropout more effectively; it achieves higher KL and MI at each rate, while maintaining modeling performance:  Negative ELBO and PPL (lower is better!) for AWD are at least as good as for the word dropout baseline or better. Both of these metrics tend to worsen at high rates, because of the lack of teacher forcing.
> >
> > Thank you for your suggestion and all your feedback throughout.
> >
> > *Note that we did remove the double-LSTM ablation to not distract from the main message. We propose to include the ablation in the appendix instead. The linked figure is preliminary, as we have not run it on all five seeds yet. (Unfortunately, we did not have all required models check-pointed, and therefore had to re-run from scratch) We propose to include this figure with updated means and standard deviations based on all five seeds in the paper upon acceptance.*

---

> > > ### Comment · Reviewer_Lidz · 2022-08-08
> > > **Re: Spot on! We revised Figure 2.**
> > >
> > > I have verified the revised Figure 2. It clearly shows that the proposed AWD significantly improves KL/MI without sacrificing PPL/ELBO compared to word dropout. I appreciate the authors for the clarification with the additional results. Lastly, I recommend authors carefully consider how to present the experimental results with their significance, as most of the reviewers initially concerned about the marginal gain. Regarding this issue, the provided arguments during the rebuttal seem to be reasonable to me. Hence, I raise my score to 6 from 5.

---

### Author Response · Authors · 2022-08-02
**Thank you very much for reviewing our paper. We made revisisions based on your feedback.**

Thank you for your useful feedback, we have made revisions to our paper (**new upload!**).

In summary, the most important changes are:
- we have included additional results on SNLI and PTB
- clarified the experimental protocol and main benefits of our approach
- added an additional reference to the theoretical analysis and fleshed out directions for future work

Detailed responses to individual reviewers are below.

---

### Meta-Review · Area_Chair_e34C · 2022-08-30

**Recommendation:** Accept
**Confidence:** Less certain

**Metareview:**

The work analyzes the use of decoder input dropout in training sequence VAEs and proposes an adversarial dropout scheme in place of the typical uniform dropout. The paper is clearly written (reviewers Lidz, WbUQ), well-motivated (all reviewers), and provides insights into posterior collapse and word dropout (Lidz, WbUQ). All reviewers were concerned that the empirical performance improvement seems small, but in light of the writing, analysis, and careful experiments I nonetheless recommend acceptance.


**Award:**

No

---

### Decision · Program_Chairs · 2022-09-14

Accept